# A disease-agnostic approach to ensemble learning for infectious disease forecasting

**Alexander C. Murph** [1,3] ✉, **Lauren J. Beesley**[1,3] ✉, **G. Casey Gibson** [1], **Lauren A. Castro** [2], **Sara Y. Del Valle**[2] **& Dave Osthus** [1]

Accurate forecasting of infectious diseases drives modern public health interventions that reduce morbidity and mortality. However, accurate forecasting in real-time remains a challenge for the modeling community. Ensembling has emerged as a critical tool for accurate forecasting by leveraging multiple component (individual) models into a single weighted average. Traditional ensembling strategies have relied on bespoke component models that weight the contributions of individual models according to extensive historical data for specific diseases. This is impractical for an emerging disease, since there would be very little – if any – data. We propose an ensembling strategy, called *epiFFORMA*, that determines component weights for an ensemble model without historical data and is therefore disease-agnostic. The epiFFORMA model builds upon the FFORMA model from the M4 forecasting competition to harness epidemiological dynamics through synthetic data. We demonstrate that epiFFORMA performs better than a naive, equal-weighting ensembling strategy when forecasting outbreaks of COVID-19, diphtheria, influenza-like illness, dengue, measles, mumps, polio, rubella, smallpox, and chikungunya. We further show that epiFFORMA, on average, performs better than the individual component models in the ensemble.

Rapid, short-term forecasts of disease spread via mathematical and computational models have been major drivers of disease intervention strategies to arrest and control modern disease outbreaks. Especially in the last two decades, disease forecasting has been the subject of focused research and public attention, and has been vital to intervention efforts such as vaccine allocation[1], social distancing[2], and other mitigation strategies[3]. The success of these and several other disease forecasting applications shows how essential real-time, accurate predictions of the behavior of an outbreak continue to be for an effective public health response.

Machine learning research has shown that ensembles of models – where predictions from several "component" (individual) models are combined into a single, weighted average – often improve predictions[4–6]. There are several meanings of the word "ensembling" within machine learning; this paper focuses on multi-model super-

ensembling[7], which considers predictions obtained from distinct models with potentially vastly different assumptions or structures (multi-model) and combines these predictions via a weighted average determined by their accuracy (super-ensemble). For example, a multi-model super-ensemble might linearly combine time series predictions from a compartmental model, a machine learning model, and a simple autoregressive model, where the weights for this linear combination are determined by a meta-model (see, for instance, ref. 8). Ensembling across models has become a popular strategy for generating more highly predictive forecasts when forecasting diseases such as influenza[9,10], dengue fever[11], respiratory syncytial virus[12], and COVID-19 forecasting[13]. The most predictive models for many disease forecasting competitions are consistently multi-model ensembles[14,15], suggesting that an informed synthesis of several models can overcome the limitations of any individual model when challenged with predicting

[1]Statistics (CAI-4), Computing and Artificial Intelligence Division, Los Alamos National Laboratory, Los Alamos, NM, USA. [2]Information Systems & Modeling (A-1), Analytics, Intelligence, & Technology Division, Los Alamos National Laboratory, Los Alamos, NM, USA. [3]These authors contributed equally: Alexander C. Murph, Lauren J. Beesley. ✉e-mail: murph@lanl.gov; lvandervort@lanl.gov

future outcomes[16]. Multi-model super-ensemble forecasters are consistently arising as the gold-standard for forecasting infectious diseases[17].

In defining a multi-model, super-ensemble forecast (which we will refer to throughout simply as an ensemble), one must define how to combine forecasts from distinct component forecast models. A common approach is to define the ensemble forecast as an equal-weighted average of individual component models, but researchers have also explored methods for estimating unequal and adaptive ensemble weights for infectious disease forecasting refs. 8,11,18,19. While these approaches have demonstrated situational success, they have their shortcomings. All approaches require sufficient training data (both historical data and/or historical forecasts) to learn the weights, making them challenging to deploy in emerging outbreak settings. Most adaptive-weighting approaches are predicated on the assumption that component models that have performed well recently will continue to perform well and should thus be up-weighted (with ref. 18 being a noted exception). Finally, weighting schemes are almost always developed and tested in a specific disease context (e.g., influenza or dengue), making their generalizability difficult to empirically assess. In addition to these challenges, the authors of ref. 20 point out that equally-weighted approaches can often perform as well or better than more complex weighting schemes in prospective forecasting settings. This is a common observation for forecasting in general: complex weighting schemes tend to perform worse than the equal-weighting scheme in empirical applications (referred to as the "Forecast Combination Puzzle" in ref. 21).

When there is no access to historical data, as is the case with emerging disease outbreaks, the current standard is to use an equal-weighting strategy and update this as data are observed[17,20]. However, there are several application scenarios where this strategy is suboptimal. For instance, whenever one or more component models are consistently the top performers and several high-variance component models are also included in the ensemble, the equal-weights scheme is the optimal method, yet also uniformly worse than the top-performing component model. This specific scenario was observed during periods of rapid change for the FluSight challenge on the 2021–22 and 2022–23 influenza seasons; the ensemble model was, on average, the best, but at any given time, there were often individual models that performed better[22]. This was also observed in retrospective forecasting of peak incidence and the week of peak incidence for the 2015–2016 Zika epidemic in Colombia[23]. This highlights another real-time advantage to equal-weighting: stability. Equal-weighted ensembles are often the safer real-time choice, while unequal-weighted ensembles have both a higher ceiling and a notably lower floor.

Our goal is to develop a method for estimating component model weights for infectious disease forecasting ensemble models that (1) is applicable to a wide range of infectious diseases, (2) is potentially deployable in emerging disease settings where little to no historical data are available, (3) does not assume recent component model success will translate to future component model success, and (4) has better performance and comparable stability to an equal-weighted ensemble (thus is as safe to use prospectively as an equal-weighted ensemble). While there are models that accomplish subsets of these objectives, to our knowledge, there are no ensemble weight estimation strategies available that can accomplish all of them. As demonstrated by the recent global SARS-CoV-2 pandemic – where an urgent, unprecedented need demanded forecasts on an emerging disease – historical data-free, disease-agnostic schemes are vital for future public health responses.

In this paper, we introduce a disease-agnostic, automated method – called *epiFFORMA* – for multi-model, super-ensemble weight estimation and forecasting that is more accurate and stable than its equal-weighted counterpart for several contemporary and historical diseases. The epiFFORMA method builds upon the FFORMA (feature-based forecast model averaging) method from ref. 24, which was one of the top-performing models from the M4 forecasting competition[25], a multi-domain, general-purpose time series forecasting competition. The core idea behind FFORMA is that different component models have strengths and weaknesses that depend on observable features of a time series and, critically, those relationships can be learned and exploited ahead of time resulting in a high-performing, unequal-weighted ensemble (this idea is similar to ref. 18). The success of FFORMA, however, depends on the availability of a large and diverse collection of time series for training. One key innovation of epiF-FORMA is that although historical data are not available for emerging outbreaks, which would typically be needed to estimate the weights in a weighted ensemble, generative outbreak models are available. Generative models can produce synthetic outbreak data that mimics the dynamics of historical outbreaks. With a sufficiently comprehensive library of synthetic data, the relationship between outbreak features and optimal component model weights can be pre-learned. When real outbreak data are then observed, weights are calculated according to a meta-model that was trained only on synthetic outbreak data, thus making epiFFORMA deployable in emerging outbreak settings. A second key innovation of epiFFORMA is the use of time series features tailored to the infectious disease forecasting context. The feature-informed weights meta-modeling can be applied whether the component model forecasts are informed by real data or not, allowing this method to be easily generalized to potentially improve infectious disease forecast super-ensembling in forecasting competitions and beyond.

We argue in this paper that epiFFORMA is an effective forecasting method for emerging diseases, as historical data are not required for training. As mentioned previously, complex weighting schemes get outperformed by a naive equal-weighting scheme in empirical applications. While it is impossible to comprehensively show that epiF-FORMA outperforms equal weighting for every possible application, we build this case by assessing the performance of epiFFORMA on 11 real-world data applications. For this paper, we focus on diseases with an airborne or vector-borne mode of transmission. Demonstrating epiFFORMA's empirical advantage over equal-weighting in additional disease settings is a key direction of future work.

Top-level information on all data used in this paper is available in Table 1. Each individual time series for a disease spans the entire time period listed. Multiple observations for a disease come from different subregions; for instance, in the US, the different subregions are typically states or territories. Within the data on a specific subregion for each disease, every possible forecast date is considered, which gives the model at least 10 weeks of data and allows for four weeks of validation data. In the following data applications, forecasts from each of these sub-timeseries are calculated one to four weeks out and compared against the truth. The one to four week forecast horizon is used for every data application in this paper and in the supplement. Furthermore, the focus of these applications is to perform forecasting (integrating over possible intervention scenarios) rather than projection (estimating counts under several conditional scenarios). More specifics on how these data are obtained and cleaned are available in the Supplementary Materials.

## Results

We compare epiFFORMA forecasts to equal-weighted and individual component model forecasts. To assess the performance of the different weighting schemes – and of the individual component models – across several data applications, we use three popular measures of performance in the disease forecasting literature[26–28]: root mean-squared error (RMSE), mean absolute error (MAE), and the weighted interval score (WIS). To perform the WIS calculation, we develop a novel method for uncertainty quantification, as discussed in the Materials and Methods Section.

**Table 1 | Metadata for every disease considered in this paper**

| Disease | Time Period | Freq. | # of Sub- regions | Mode of Transmission |
|---|---|---|---|---|
| COVID-19 (US) | 2020–09/2023 | Weekly | 56 | Airborne droplet |
| COVID-19 (Global) | 2020–09/2023 | Weekly | 193 | Airborne droplet |
| Influenza-like Illness (US) | 1997/2010–2023 | Weekly | 75 | Airborne droplet |
| Dengue Fever (Global) | 1995–2023 | Weekly | 25 | Vector-Borne (Mosquitos) |
| Diphtheria (US) | 1916–1948 | Monthly* | 32 | Airborne droplet |
| Measles (US) | 1928–2002 | Monthly* | 51 | Airborne droplet |
| Mumps (US) | 1968–2003 | Monthly* | 32 | Airborne droplet |
| Polio (US) | 1928–1968 | Monthly* | 32 | Airborne droplet/ Fecal-Oral |
| Rubella (US) | 1966-1999 | Monthly* | 22 | Airborne droplet |
| Smallpox (US) | 1928–1948 | Monthly* | 16 | Airborne droplet |
| Chikungunya (Brazil) | 2013–2022 | Weekly | 27 | Vector-Borne (Mosquitos) |

Specific information on cleaning procedures and data sources can be found in the Supplementary Materials.
*Aggregated up from weekly to 4-week data.

The component models used in the ensemble are chosen for speed of computation, as they must be fit in real-time on new data observations, and for overall prevalence in the field of disease forecasting. Some of these models are implemented using the `forecast` package in R[29]; the models implemented from this package are:

1. rw, random walk forecast, with drift = FALSE;
2. theta, equivalent to simple exponential smoothing with drift (see ref. [30]);
3. arima, autoregressive integrated moving average;

We also included several models not implemented via the forecasting package:

4. gam, generalized additive models[31];
5. moa, method of analogs[32], using synthetic data[33];
6. moa-deriv, the moa method on the derivative scale;
7. meanfcst, mean of the time series for the two years immediately preceding the forecast date. When less than two years are available, all available data are used;
8. mirror, predictions symmetric about the start of a forecast, out through the forecast horizon (e.g., next week's forecast equals last week's observation);
9. gam2mirror, a linear combination of the gam and the mirror models, with more weight placed on the mirror model as the forecast horizon grows.

For further details on the specific implementations of these component models, and a discussion on the models themselves, see the Supplementary Materials. We also evaluate a forecast model that is an equally weighted average of the above component models, excluding the mirror model.

### EpiFFORMA up-weights component models that give more accurate forecasts

We begin with an example of an epiFFORMA forecast to show how it up-weights component models that perform better in specific scenarios. In Fig. 1a, COVID case counts in California, USA, are plotted from July 2020 up through September 2020. The 1-week through 4-week ahead forecasts from late August onward are given for each component model and for epiFFORMA. In Fig. 1b, the weight epiFFORMA assigns to each component model is displayed for each forecast horizon. Note that the moa, meanfcst, and moa_deriv models have the strongest agreement with the behavior of the true data values. Correspondingly, epiFFORMA gives these three models the most

weight over all of the forecast horizons. The moa_deriv model is close to the true data value for the first forecast horizon and gets progressively worse with a growing forecast horizon. For this component model, the weighting scheme in epiFFORMA performs as expected: the weight assigned to moa_deriv is decreasing with increasing forecast horizon. In this single example, epiFFORMA is functioning as desired: more accurate component models are up-weighted while less accurate ones are down-weighted. This example is representative of epiFFORMA's behavior. It consistently up-weights (down-weights) better (worse) performing component models on a context-specific basis. Further evidence of this is shown in Supplementary Fig. 1.

### EpiFFORMA outperforms a naive equal-weighting scheme

The forecasting accuracy of epiFFORMA in terms of MAE, RMSE, and WIS is, on average, significantly better than the equal-weight ensemble. In Fig. 2, the average RMSE and MAE are plotted for every disease in Table 1, normalized according to the corresponding average error value for the equal-weight ensemble (excluding the mirror model). Compared to all the individual component models plotted, only the values for epiFFORMA are consistently outperforming the equal-weights ensemble, i.e., to the left of the line "value = 1". On average, epiFFORMA's MAE and RMSE are 87.9% and 93.7% of those of the equal-weighted ensemble, respectively. In terms of overall averaged MAE and RMSE for each individual disease, epiFFORMA is only outperformed by the equal-weighting scheme when forecasting mumps (and in this case, just barely). In terms of overall averaged WIS for each individual disease, epiFFORMA outperforms the equal-weighting scheme for every disease (see Supplementary Materials).

### EpiFFORMA out-ranks a naive equal-weighting scheme

In terms of the rankings for accuracy of forecasts, epiFFORMA, on average, outperforms the equal-weight ensemble and the individual component models across a majority of the empirical applications. In Fig. 3, the average performances for epiFFORMA, the equal-weight ensemble, and the individual component models are ranked (from smallest to largest) in terms of average forecasting error. In terms of both MAE and RMSE, epiFFORMA outperformed the equal-weight ensemble on each of the 11 real-world data applications except for mumps and (in the case of RMSE) chikungunya. Even in scenarios where it shows worse average performance in terms of either MAE or RMSE, epiFFORMA's error value is extremely close to whichever model outperformed it, and there are only three instances where epiFFORMA was outperformed in terms of both error measures.

Especially for RMSE, epiFFORMA occasionally performs worse than some of the individual component models. However, as is

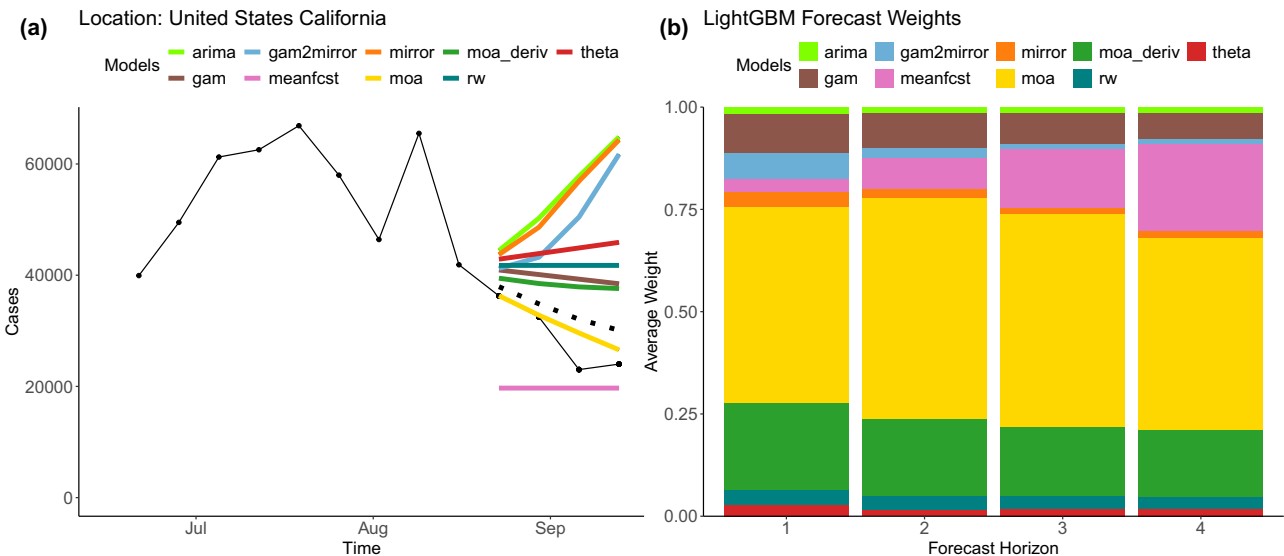

**Fig. 1 | EpiFFORMA forecasts of COVID-19 in California.** (left) epiFFORMA's forecast (in the dotted black line) and the component model's forecasts (in the colored lines) for cases of COVID-19 in California. The solid black line indicates the observed case counts. The epiFFORMA model forecasts downward, which coincides with the true values. (right) The weights assigned to each component model in epiFFORMA for each horizon. The three best individual component models (moa, moa_deriv, and meanfcst) are all assigned the most weight, though the distribution of those weights changes as a function of forecast horizon.

outlined in Fig. 2, epiFFORMA is more consistently a top-performing model. In fact, the worst epiFFORMA ever performs across all 26 scenarios (13 diseases and 2 error measures) is third, and that only happened once. The best minimum rank across the other 10 models belongs to the theta model, whose worst rank is 7th. Furthermore, the average epiFFORMA rank across the 22 scenarios is 1.45 (1st 13 times, 2nd 8 times, 3rd 1 time). The next best-performing model is the equal_wt with an average rank of 3.27. This means epiFFORMA's worst rank is better than equal_wt's average rank. All assessments thus far of epiFFORMA have been in terms of average/median performance over several diseases and several forecast horizons. We visualize the entire distribution of forecast rankings of epiFFORMA and compare this to the forecast rankings of the equal-weight ensemble in terms of MAE in the Supplementary Materials.

## Discussion

When faced with an emerging epidemic for which there are no historical data, ensembling methods typically rely on a naive equal-weighting scheme to determine component weights. Especially at the onset of a disease crisis, where reliable forecasts are paramount to public health responses, urgent need demands more accurate forecasts and more sophisticated methods. EpiFFORMA is an ensembling framework demonstrated to perform better than the equal-weighting scheme on a large number of empirical applications. Evidence on these applications shows how epiFFORMA overcomes the Forecast Combination Puzzle[21,34], where more sophisticated methods for ensemble weighting are typically outperformed by the equal-weighting scheme, which is a phenomenon that continues to be observed in forecasting applications today[17,20].

Nine forecasting models are used in an ensemble to calculate forecasts for 11 different historical disease applications. In terms of averaged MAE and RMSE, and median WIS, the epiFFORMA model performed better than the equal-weighting and the individual component models for a strong majority of the applications. In scenarios where epiFFORMA is outperformed by another model, the difference is small. See the Supplementary Materials for WIS results.

EpiFFORMA outperforms equal weights because the GBM weight model is able to identify which models do well (or poorly)

under what circumstances. For instance, Fig. 1 shows that the best individual model is MOA, and the meta model is able to assign the largest weight to MOA. In the Supplementary Materials, we investigate further how the meta model is able to predict which component models will do best and worst and assign weights to those models accordingly.

To analyze this further, we performed a simple cluster analysis on the weights (in our case, vectors of length 9). The clusters that are roughly closest to equal weights contain roughly 40.27% of the weights produced by the epiFFORMA ensembling meta-model across every forecast application in this paper; roughly 59.73% of the time, epiFFORMA produced a distinctly non-equal weights profile. This shows that the unequal weight model is not always a better choice; what unlocks the Forecast Combination Puzzle is having a meta model able to discriminate between when an equal weight model is best and when something else is likely to be better.

One general observation on the epiFFORMA method is that it performs best directly after a peak is observed, and right when cases begin to surge at the start of an outbreak. In these scenarios, it down-weights models that perform poorly at peaks, and up-weights models that perform best in that specific scenario. In other regimes, such as prior to a peak, epiFFORMA performs more similarly to the equal-weight ensemble. A further analysis on the performance of epiFFORMA partitioned by phases of a disease outbreak (as defined according to the shapelet-based methods of ref. 35) is in the Supplementary Materials.

While the implementation of epiFFORMA used for these applications shows strong empirical performance, there are several directions for further development. Since epiFFORMA is an ensembling framework, it is possible to use this framework on any set of component models, as long as these models can provide forecasts in real-time. While we focused on evaluating readily available models, more sophisticated component models can and should be considered. We have demonstrated that epiFFORMA is capable of learning what scenarios each component model does and does not perform well. While the component models used in this paper are relatively simple, they demonstrate epiFFORMA's capabilities to pre-learn these scenarios. Future work should explore the potential for epiFFORMA to

outperform gold standard equal weights ensemble forecasters in, for example, infectious disease forecasting competition benchmarking.

As discussed further in the next section, epiFFORMA pre-learns weights on an expansive library of synthetic data. Intuitively, the more variation represented in this library, the more robust the component

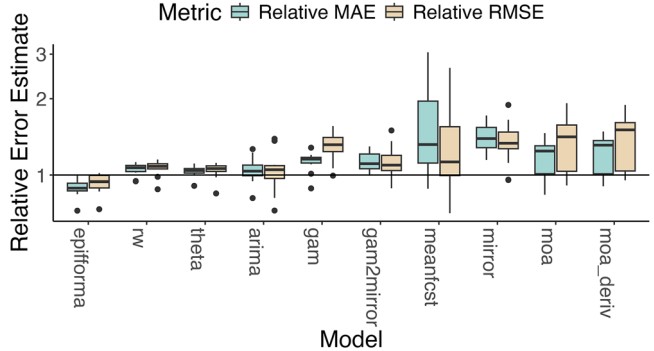

**Fig. 2 | Performance of epiFFORMA and the individual component models compared to the equal-weighting scheme.** This equal weighting scheme placed equal weight on all component models except the mirror model, which received no weight. This was because the mirror model is only a reasonable model to forecast outbreaks at their peaks and is unlikely to be a good forecasting model during other phases of an outbreak, or in general. Boxplots summarize the average performance for each of the diseases outlined in Table 1 (n = 11) using MAE and RMSE as the measures of error. The values visualized by the boxplots are (in order from top to bottom) maximum (sans outliers), 75th quantile, median, 25th quantile, and minimum (sans outliers). Outliers, determined by 1.5 times the interquartile range, are visualized as black points. Every average performance value for a given disease is divided by the corresponding value for the equal-weights ensemble; observations below the line value = 1 correspond to instances where a model outperformed the equal-weights ensemble.

weights model should be. How well the synthetic data represent real disease dynamics is analyzed via the Uniform Manifold Approximation and Projection method from[36] in the Supplementary Materials. There also exists a broad range of further developments with regard to how a given time series is mapped to a set of optimal weights, which we describe further in the Materials and Methods Section.

There are several directions for future work beyond additional development on the implementation of epiFFORMA. One direction is to investigate epiFFORMA for multivariate time series, such as applications where the interest is forecasting both cases and deaths. The authors also intend to improve upon the Uncertainty Quantification methods developed in this paper, using methods such as quantile regression or conformal prediction adjustments to get closer-to-nominal coverages.

While we view the historical data-free nature of this method as its main strength, nothing precludes us from augmenting the synthetic library with historical outbreak data. Including cross-pathogen and cross-surveillance system data streams for forecasting is an emerging interest in the literature, especially in the context of the "throw more data at it" mentality brought forth by advances in AI. See, for instance, the Flusion model[37], which placed 1st in the 2023–24 FluSight competition[38], and several other recent papers[39–41]. Although this paper investigates the scenario where only synthetic data are available, we will investigate including different historical data sets in an upcoming paper.

## Methods
EpiFFORMA uses a library of synthetic data to pre-learn the weights of an ensemble model under every scenario represented in the library. This is achieved by transforming each synthetic time series into a vector of time series features (i.e., summaries of the time series), finding the optimum combination of pre-specified component models for that time series, and learning a meta-model between the feature

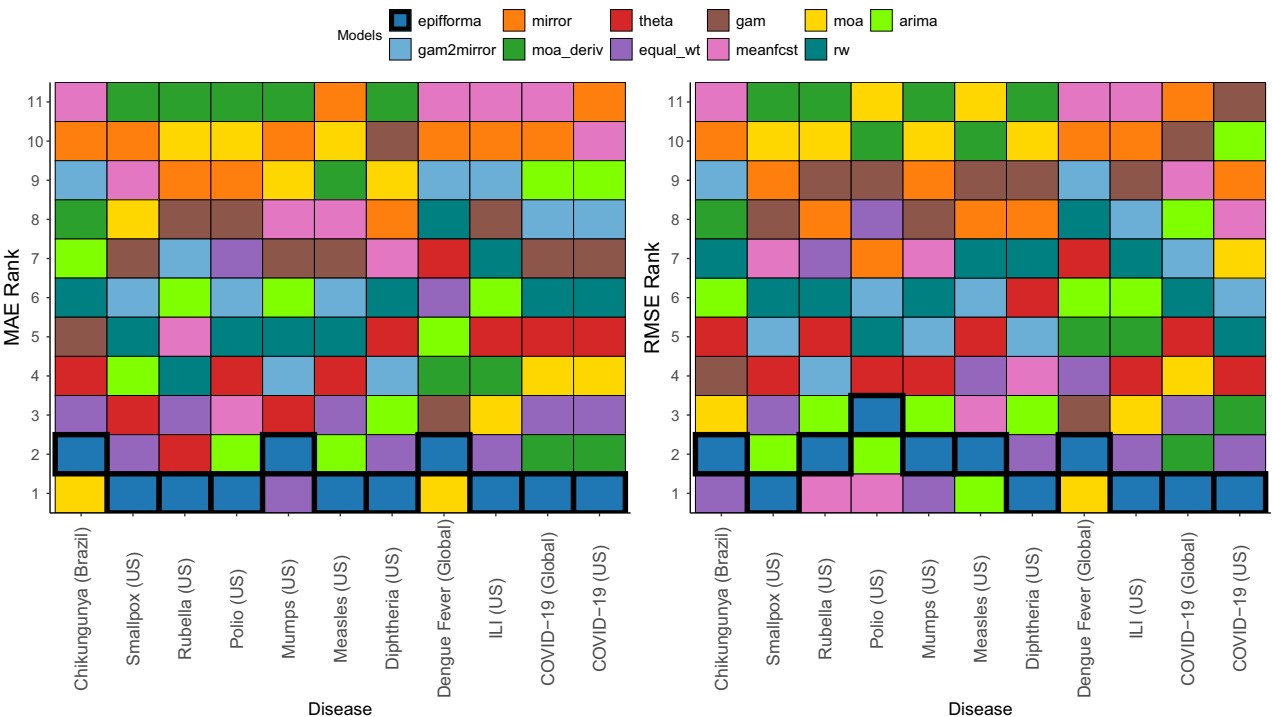

**Fig. 3 | The rank of epiFFORMA compared to the equal-weighting scheme and the individual component models.** This figure considers all the diseases in Table 1. The overall average performance for each model is ranked (along the vertical axis), for each disease (along the horizontal axis). Both in terms of MAE and RMSE, epiFFORMA is consistently the top-performing model. The equal weighting scheme evaluated here excluded the mirror component model.

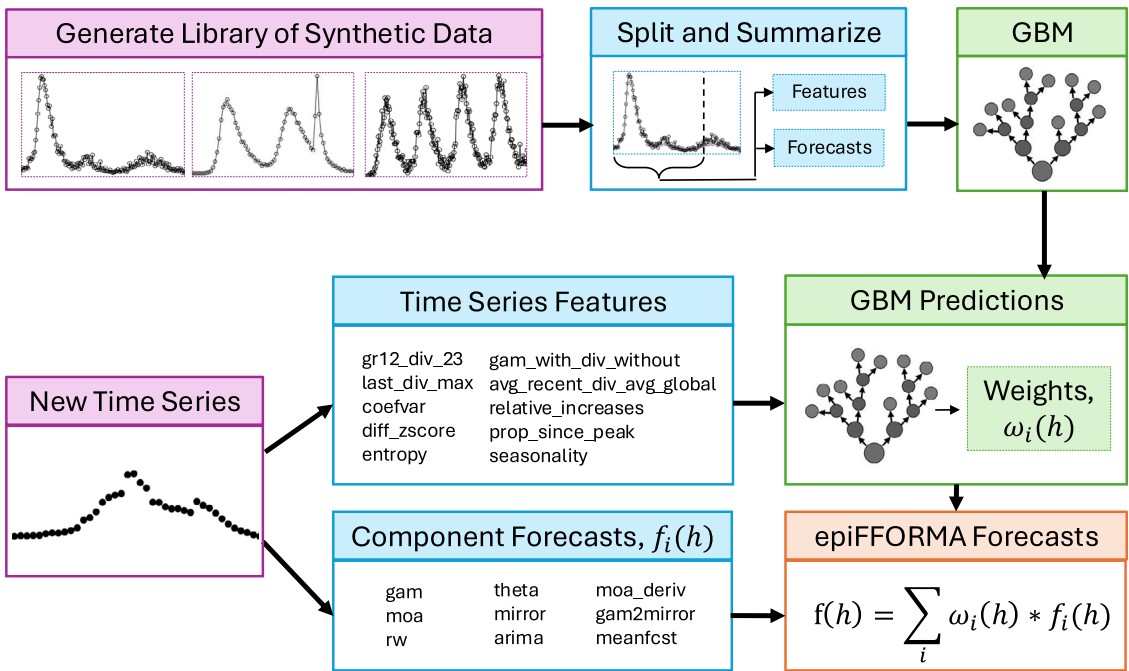

**Fig. 4 | Outline of the epiFFORMA method.** This outline uses the models and features discussed in the Implementation of epiFFORMA Section. Prior to real data being observed, the Gradient Boosting Machine (GBM) is fit according to forecasts made on data generated by the Generative Models. When a new time series is observed, the time series features are calculated, and forecasts are made by each component model. Using the time series features, weights are estimated using the pre-fit GBM, and these weights are then used to average together the component model forecasts. Compare this formulation of FFORMA, found in ref. 52.

vector and optimal weighting scheme. When a new time series is then observed, its feature vector is extracted and passed into the weights meta-model, which outputs the ensemble model weights. This entire process is outlined in Fig. 4. The offline process for pre-learning the weights is visualized at the top: the synthetic data are created by a class of generative models, forecasts are performed for each synthetic datum using the component models and assessed according to a known truth, then the time series features calculated for these time series are used to fit a meta-model – which for this implementation is a Gradient Boosting Machine (GBM) – that predicts the best-performing component model based on time series features. Our GBM approach to ensemble weight estimation itself relies on ensembling through boosting and averaging across multiple GBM model fit replicates. We clarify that by "ensemble" we are referring to the weighted combination of the distinct component models rather than the various levels of ensembling used in the weights estimation meta-model.

At the bottom of Fig. 4, we visualize the epiFFORMA method once a new time series is observed. Much like the process for the synthetic data, both time series features and component forecasts are calculated for a new data point. Using the features, the pre-fit meta-model is used to determine the component weights, which are then used to weight the ensemble for the final prediction.

The epiFFORMA method learns a set of weights that should be interpreted differently from those found in a traditional ensembling scheme. Suppose there are forecasts from $K$ component models, and suppose each model produces an $h$-step ahead forecast for an infectious disease time series $\{y_1, ..., y_t\}$, for some positive integer $t$. We denote the $h$-th step ahead forecast for component model $k$ ($k \in \{1, ..., K\}$), fit on the first $t$ timepoints, as $\widehat{Y}_k(h, t)$. A traditional ensembling framework linearly combines these forecasts,

$$\widehat{Y}(h, t) = \sum_{k=1}^{K} w_{k,h} \cdot \widehat{Y}_k(h, t), \qquad (1)$$

where $w_{k,h}$ is the ensemble weight for component model $k$ and the $h$th horizon. In epiFFORMA, weights are assigned according to the order of component forecasts. That is, let $\widehat{Y}_{[k]}(h, t)$ be the ordered $h$th step ahead forecasts for the component models such that

$$\widehat{Y}_{[1]}(h, t) \leq \widehat{Y}_{[2]}(h, t) \leq \cdots \leq \widehat{Y}_{[K]}(h, t).$$

The ensembling approach in epiFFORMA can then be written,

$$\widehat{Y}(h, t) = \sum_{k=1}^{K} w_{[k],h} \cdot \widehat{Y}_{[k]}(h, t), \qquad (2)$$

where $w_{[k],h}$ is now the ensemble weight for $k$th component model at the $h$th horizon, ordered according to the value predicted.

The underlying intuition in predicting the weights for the order of models is that it is expected to be more stable than predicting the weights for each component model specifically. Rather than learning when specific models need to be up-weighted or down-weighted, epiFFORMA learns under what circumstances model predictions at the fringes should be up-weighted and under what circumstances the central tendency of the model predictions should be up-weighted. Furthermore, when one component model gives a particularly incorrect prediction, epiFFORMA does not need to rely on the GBM to down-weight that specific model, but instead identifies scenarios where models that are egregiously high or low should be down-weighted. While there is not necessarily a rigorous theoretical basis for this intuition, experiments both on synthetic and real data have shown that predicting weights on the order of models improved the overall performance of epiFFORMA.

As mentioned previously, the weights $\{w_{[1],h}, ..., w_{[K],h}\}$ are modeled as a function of features derived from the time series:

$$\mathcal{F}(\{y_1, \ldots, y_t\}) = \{\text{set of features of series}\}$$
$$\mathcal{G} \circ \mathcal{F}(\{y_1, \ldots, y_t\}) = \left( w_{[1],h} \ldots w_{[K],h} \right). \quad (3)$$

In epiFFORMA, a GBM for the function $\mathcal{G}$ is used, fit using the `lightGBM` package in R[42]. This specific non-linear regression model is used due to its fast prediction time, its ability to handle multiple features of varying data types, and because it only requires the optimization of a relatively small number of parameters. This method was used by every contestant ranked in the top 50 in the M5 forecasting competition, and has become a standard for several other forecasting applications (see ref. [43] for more details on the prevalence of `lightGBM`).

The synthetic data used to pre-learn the GBM consists of 60,000 time series. Given a desired forecast horizon $h$, all time series are split into training and testing periods according to a randomly selected split point between the 11th and $(T-h)$th timepoints, where $T$ is the total length of the time series. The probability of each split point is weighted to be proportional to the absolute value of the square root of the rate of change at that point. Via resampling of these time series (where a resampled time series may be given a different training and testing period), 180,000 training and testing periods are created.

The time series features in (3) are calculated, and the component models in (2) are fit using the training period only. Forecasts are then generated for the test period for each synthetic datum, and the forecasting errors are calculated. The GBM loss function minimizes the averaged forecasting errors with respect to the component weights in (2) as a function of the features derived from the training period. The specific metric used to determine forecasting errors and the loss function used for the GBM are discussed below.

While epiFFORMA is heavily inspired by the FFORMA method, there are several, fundamental differences. The first and most significant difference is that epiFFORMA has been developed specifically for forecasting emerging epidemics. While FFORMA was trained on 100,000 general time series provided by the M4 forecasting competition, epiFFORMA is specifically trained on synthetic data meant to emulate the behavior of a broad class of disease outbreaks. In this way, epiFFORMA is truly agnostic to real data, since it is only pre-fit to synthetic data, while the FFORMA method is pre-fit using real data. Furthermore, the features learned from the time series and the component models in the ensemble are catered to the behaviors of outbreaks. For instance, the mirror model outlined in the Results Section was proposed as a reasonable model to forecast outbreaks at their peaks, but is unlikely to be a good forecasting model during other phases of an outbreak, or in general. For that reason, the mirror model was not included in the calculation of the equal weights forecasts.

Two other distinctions between FFORMA and epiFFORMA are that epiFFORMA weights the order of the component models (as outlined above) instead of weighting the models themselves, and epiFFORMA uses `lightGBM` to fit the Gradient Boosting Machine (not `xgboost`), which was a method identified as superior for forecasting in the M5 competition[43].

## Uncertainty quantification

Prediction intervals on a given forecast are calculated using the prediction intervals produced by the arima, theta, and gam models. The width of the epiFFORMA prediction interval—assumed symmetric around the epiFFORMA forecast—is computed as a weighted combination of the widths of the three prediction intervals. The weights for this linear combination are determined by a second epiFFORMA metamodel fit on a synthetic data library that is separate from the one used to fit the original ensemble.

## Implementation of epiFFORMA

As epiFFORMA is a general method for performing data-free forecast ensembling on emerging epidemics, the specific implementation can vary between applications. In the following, we give an overview of the implementation developed for the data applications in this paper. The model names in this section reference those found in Fig. 4, which is meant to outline this specific implementation as well as give a general overview of the epiFFORMA method. While we show in the Results Section that this implementation has impressive performance on real data, we do not claim that this implementation is optimal. Improving upon the implementation discussed here is an important direction for future work.

**Generative models.** An underlying assumption of epiFFORMA is that the dynamics represented in the synthetic data cover the dynamics of real disease outbreaks. While no individual generative model is expected to represent all disease spreads, many different scenarios for the progression of a disease will be represented with a sufficiently comprehensive library of synthetic data. When the dynamics (according to the time series features) of a new data observation are well-represented by some behavior found in the synthetic data, we propose that the weighting scheme that performed best for this artifact will also perform best on the new data observation.

Assessing how well the synthetic data cover the dynamics of real disease outbreaks can be done via the Uniform Manifold Approximation and Projection (UMAP) method of ref. [36]. Such an analysis is undertaken in the supplementary materials.

While we stress that epiFFORMA is data-agnostic, developing a synthetic library of data that well-represents the many different scenarios for the progression of a disease outbreak requires at least a cursory sense of what real-world outbreaks "look like." While encoding this intuition into the synthetic data may qualify as incorporating historical data into the model, we argue that this model remains data-agnostic for several reasons. First, no real-world time series are included in the synthetic library; all time series generated are variations on a Susceptible-Infected-Recovered (SIR) model. Second, under each parameterized mechanism for producing a time series (discussed below), a wide range of possible parameters is randomly generated for the purpose of producing a wide amount of variation, not for producing a specific type of variation. Lastly, each mechanism developed below is derived from methods meant to model a disease outbreak, meaning that philosophically, they encode a theory on how diseases are expected to behave. The synthetic library is, therefore, a mechanism to incorporate several different theories on modeling diseases into a single machine learning model. Of course, these expectations might also be driven by real-world expectations, but not always. For instance, scientists were interested in modeling multi-wave type behaviors before these behaviors were actually observed during the COVID-19 pandemic[44,45].

With the aim of creating a synthetic data library with widely varying time series features, we primarily use the differential equation view of the SIR model to simulate a variety of epidemic curves (many of these calculations were done via the `deSolve` package in R)[46]. We use 3 different strategies for creating synthetic data, sampling 20,000 curves from each model under random initial conditions.

The first strategy for creating synthetic data used for epiFFORMA is the generation of time series via mixtures of SIR curves. We randomly generate between 3 and 7 SIR curves with random initial conditions, and order the peaks so that they are increasing in time, decreasing in time, or totally random. This generation strategy leads to a "roller coaster" type curves. The second strategy for creating synthetic data is the same as the first strategy, but an additional sinusoidal wave is added to the entire curve. This results in "roller coaster" type curves that also have local variation in a periodic fashion. The last strategy for creating synthetic data is designed to emulate different

seasonal effects. Waves are randomly sampled from several SIR curves, then each is spaced 52 weeks apart.

Further details on each generative model are found in the Supplementary Materials. We also provide a discussion of the representation of the synthetic data on real disease data in terms of a simple metric and using a Uniform Manifold Approximation and Projection technique[36].

**Time series features.** We extract features from the time series using features we believe are relevant for epidemiological forecasting. The aim in choosing this set of features is to comprehensively encode all the dynamics present in the data without overburdening the meta-model with too many inputs. After several rounds of experimentation, we determined that the following features led to good empirical performance for epiFFORMA:

1. gr12_div_23, expression of the rate of change of the time series in the recent past;
2. last_div_max, how the last value compares to the maximum and minimum of the last 10 values;
3. coefvar, coefficient of variation on the last 10 values;
4. diff_zscore, how large is the jump between the last two values compared to the past two years;
5. entropy, spectral entropy calculation from the `tsfeatures` package[47];
6. gam_with_div_without, by how much do the GAM predictions change when the last point is and is not included;
7. avg_recent_div_avg_global, mean of the last 10 values over mean of the last two years of values;
8. relative_increases, what is the current consecutive increase, and how does this compare to the maximum number of consecutive increases;
9. prop_since_peak, distance in time from period with the largest average data value (within seasons);
10. seasonality, maximum autocorrelation across several possible lags.

For more specific details on these features, see the Supplementary Materials.

**GBM parameters and loss Function.** As mentioned previously, the GBM used as a meta-model for the epiFFORMA method is the one developed in the `lightGBM` package[42]. This decision was made because of `lightGBM`'s recent and spectacular successes reported in the forecasting literature (see, for instance,[43]). This recent popularity of `lightGBM` may be in part due to its faster computation times for larger datasets. As is the case with many machine learning algorithms, there are several tuning hyperparameters that must be determined a priori, and an objective function must be specified according to the application. The GBM needed for epiFFORMA takes in the set of time series features and outputs numeric values for each category of the response, which in this case is the order of the component models. We used a softmax objective function[48] on these numeric values to normalize them to be probabilities, and a cross-entropy loss function (otherwise known as the Multinomial logistic loss) to be used as an objective when fitting the GBM[49].

Tuning parameters within the `lightGBM` package were given their default values if they were deemed not to affect over-fitting. To select the parameters that do affect over-fitting, we used a Bayesian Optimization scheme to predict the optimal hyperparameter settings on a hold-out set of synthetic data[33,50]. The parameters within the `lightGBM` R package fit using this method were num_leaves, learning_rate, feature_fraction, max_depth, early_stopping_rounds, and prop_holdout. For a detailed description of each of these, see the documentation found at ref. 51.

## Reporting summary

Further information on research design is available in the Nature Portfolio Reporting Summary linked to this article.

## Data availability

All data used in this study were publicly available at the time of access. COVID-19 case data were obtained from the Johns Hopkins University Center for Systems Science and Engineering GitHub repository (https://github.com/CSSEGISandData/COVID-19_Unified-Dataset), accessed September 20, 2023, and include reported cases between January 2, 2020, and March 31, 2023. Influenza-like illness (ILI) data were downloaded from the U.S. Centers for Disease Control and Prevention FluView portal (https://gis.cdc.gov/grasp/fluview/fluportaldashboard.html), accessed on or before January 31, 2024. Dengue fever data were obtained from the NOAA Dengue Forecasting Project (https://dengueforecasting.noaa.gov) and OpenDengue (https://opendengue.org), accessed May 14, 2024. Chikungunya case data for Brazil (2013–2022) were obtained from https://github.com/wmarciel/Chikungunya-in-Brazil-2013-2022, originally sourced from the Brazilian Ministry of Health. Diphtheria, measles, mumps, polio, rubella, and smallpox data were obtained from Project Tycho (https://www.tycho.pitt.edu/data/#datasets). All data were processed as described in the Methods section, including aggregation to weekly or four-week cadences where applicable, exclusion of locations with insufficient reporting, truncation of leading and trailing excess zeros, and treatment of rare negative case counts. No new datasets were generated during the current study.

## Code availability

All codes and data used in this paper are available in a Zenodo repository: https://zenodo.org/records/16582447.

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

## Acknowledgements

Research presented in this article was partially supported by the Laboratory Directed Research and Development program of Los Alamos National Laboratory under project number 20240066DR, awarded to S.Y.D.V. (PI) and D.O. (co-PI). Los Alamos National Laboratory is operated by Triad National Security, LLC, for the National Nuclear Security Administration of the U.S. Department of Energy (Contract no. 89233218CNA000001). This research was partially funded by NIH/NIGMS under grant R01GM130668-01 awarded to S.Y.D.V.

## Author contributions

A.C.M. and L.J.B. contributed equally to this work. D.O. conceived the study. L.J.B. curated the data. A.C.M., L.J.B., G.C.G., L.A.C., and D.O. developed the methodology, performed the analyses, and conducted the modeling. A.C.M., L.J.B., G.C.G., L.A.C., and D.O. implemented the software and carried out validation. A.C.M. and L.J.B. wrote the original draft of the manuscript. D.O., L.A.C., and S.Y.D.V. contributed to the review and editing. S.Y.D.V. and D.O. supervised the project. S.Y.D.V. acquired funding. All authors approved the final manuscript.

## Competing interests

The authors declare no competing interests.
