## [Transparent Peer Review file · Nature Communications]

A Disease-agnostic Approach to Ensemble Learning for Infectious Disease Forecasting

Corresponding Author: Dr Alexander Murph

Version 0:

Reviewer comments:

Reviewer #1

(Remarks to the Author)

Overview

This was a really interesting manuscript and was well written. Framing their methodology in the lens of the forecast combination puzzle is clever and make sense. I think a broad audience can appreciate a paper like this, but generally I think some improvements needs to be made to make the manuscript more approachable, including added rationale in some instances.

Major comments

- In the instances when epiFFORMA did not perform the best, do you know why? Was there a certain feature of the outbreak that it could not match? I think more discussion about the pros and cons of the weighting scheme in reference to the characteristics of outbreaks could improve the discussion.
- Did you consider incorporating looking at forecasting targets as an outcome (e.g., peak week, incidence at peak week)? This could provide a more holistic picture of the weighting scheme relative to other models.
- I thought it was an interesting choice to only train the model on synthetic outbreaks rather than any real outbreaks. Given your central argument is that the tool is disease agnostic, you could just include outbreaks from all different types of settings and pathogens. In that sense, you could additionally be accounting for (or at least considering) the differences in reporting and disease surveillance, which in the current method of synthetic outbreaks, is not accounted for (or at least it wasn't clear to me how it was accounted for). Assuming you continue forward with the current method of training the models only on synthetic data, I think it warrants a bit more rationale and discussion as to why this choice was made (presumably related to the number of time series available to train the model on, but I am not sure). And please comment further how reporting (is it time-varying?) is accounted for in the synthetic outbreaks.

Minor comments

- In the abstract, ILI has not yet been defined and may not be a well-known term to folks outside of infectious disease modeling. (also in Table 1, please define ILI)
- 'Dengue' does not need to be capitalized
- Page 2, 'forecaster' should be forecast, I think
- Page 2, third paragraph – this situation was also observed with Zika virus (<https://www.nature.com/articles/s41467-021-25695-0>)
- The introduction reads very nicely
- Page 3, the data truncation process is not clear, please add more details
- Page 4, you refer to the top and bottom of Figure 1, I think you mean left and right? In any case, likely would make more sense to add A and B designations so you can be more precise in the reference to the figure
- Figure 2 – why did the mirror model receive no weight? (this is brought up on page 10, but there should be a footnote in this figure to clarify)
- Figure 2 – I would suggest changing the y-axis label to something more informative... maybe "Relative error estimate" or something
- Page 7 – I don't think you've defined 'UQ'
- Figure 4 – Please define GBM in the caption
- It is useful to include the underlying intuition section. Consider adding a similar description for what the order represents.

(Remarks on code availability)

Reviewer #2

(Remarks to the Author)

This paper presents an adaptation of the FFORMA ensemble forecasting approach, applied to forecasts of epidemic dynamics (case incidence over time). In my understanding, FFORMA produces an ML meta model (here a gradient-boosted machine) that maps expert-selected timeseries features to vectors of weights over an ensemble of forecasting models. The intention is to produce a fine-tuned weighting of the different ensemble members based on how well they are expected to perform given the salient characteristics of the timeseries used as input to the forecast.

The novel contribution of this paper is a domain-specific application of the FFORMA approach that the authors call epiFFORMA, where the epi prefix refers to the application to epidemiological dynamics as the forecast target. This domain-specific application manifests in two major ways: the first is that the timeseries features used to train the FFORMA GBM are selected based on general expert knowledge of epidemic dynamics; the second is that the GBM was trained using a wide array of synthetic data produced by mechanistic simulation models of epidemic dynamics.

The reason synthetic data was used rather than real data was because a major objective is to produce a tool that can be used in the context of an outbreak of an emerging pathogen for which historical data does not exist. This is certainly an important objective, and this paper presents a useful step towards it.

After reading the paper carefully, my overall opinion is that while the research presented is important and has a lot of potential, the status of these results is very much a 'proof of concept' and lacks a convincing demonstration that the approach is state-of-the-art. Indeed, the authors note this in their discussion regarding future studies: "Future work should explore the potential for epiFFORMA to outperform gold standard equal weights ensemble forecasters in, for example, infectious disease forecasting competition benchmarking."

From a scientific perspective, I found the paper somewhat unsatisfying because it presents empirical comparisons without explaining them. This was especially true with respect to the discussion about the Forecast Combination Puzzle. While it's clear that under the conditions presented in the results the epiFFORMA approach out-competed equal weights under most circumstances, it's not clear that this is unique to the epiFFORMA approach (other weighting techniques were not demonstrated for comparison). More importantly, the result was not explained in terms of the mathematics and statistics underlying the forecasting process - I was left wondering "why does this method out-perform equal weights?" an explanation would allow the result to be understood in the context of the literature around the FCP, and could potentially advance the field.

If given the opportunity to revise, I suggest a making a major effort to demonstrate this method relative to state-of-the-art epidemic forecasting techniques, particularly if those use equal-weighting and a contribution can be made towards explaining the FCP in this context.

In summary:

strengths: highly-relevant topic; promising proof-of-concept; rigorous methodology applied so far.

weaknesses: empirical observations without scientific explanation or generalisation; limited proof-of-concept only - demonstration is not state-of-the-art and does not attempt to approach it.

(Remarks on code availability)

The db appears to contain legacy or non publication-related code (e.g., `stuff_for_castro`) I recommend making an updated user-facing release. The Zenodo database was a single 3GB zip file which prevented me from downloading and testing any of the code. Also it did not appear to be linked to an active repository.

Version 1:

Reviewer comments:

Reviewer #1

(Remarks to the Author)

Thanks for addressing my comments (and the other reviewer's comments) and identifying when suggestions were out-of-scope. The manuscript reads very well and I think it will be an important contribution to the field. I'm excited to see the group's upcoming work on cross-pathogen forecasting as well.

No other comments from me.

(Remarks on code availability)

I did not assess the code.

Reviewer #2

(Remarks to the Author)

Thank you for considering and responding to my comments. I have no further suggestions.

(Remarks on code availability)

I checked to make sure the code was accessible in the repository linked above. The issues noted in my previous review have been addressed.

Responses to Reviewer Comments on Manuscript “Beyond Equal Weights: A Disease-agnostic Approach to Ensemble Learning for Infectious Disease Forecasting” for Nature Communications

May 2024

1 Note to all Reviewers:

We would like to thank all the reviewers and the Editor for their diligent suggestions for our article. Below, we have enumerated all suggestions and given our responses in blue.

2 Reviewer 1 Comments:

Major Comments

1. In the instances when epiFFORMA did not perform the best, do you know why? Was there a certain feature of the outbreak that it could not match? I think more discussion about the pros and cons of the weighting scheme in reference to the characteristics of outbreaks could improve the discussion.
 - Response: *Thank you for this comment. In the supplementary materials (Section 6), we have analyzed the performance of epiFFORMA in terms of MAE and RMSE broken up according to different disease outbreak behaviors: the initial surge, the period of increase, before and after the peak, the descent of cases, and the flattening out of cases. We found that epiFFORMA typically shows performance gains during the surge, post-peak, descent, and flattening out, and occasionally gets out-performed by equal weights during the ascent and pre-peak (though in these sections they are still mostly comparable). With regards to epiFFORMA (and, by extension, the synthetic data) not being able to match certain features of the outbreak, this is the fundamental question about the synthetic data, and an ongoing interest for our research team. Our UMAP analysis explores this question by mapping the time series to a space where they can be visually compared in two dimensions. We found that while some disease dynamics are not well caught by the synthetic data, the coverage is very strong; this is exhibited practically by our applications showing an improvement over equal-weights when epiFFORMA is trained on synthetic.*
2. Did you consider incorporating looking at forecasting targets as an outcome (e.g., peak week, incidence at peak week)? This could provide a more holistic picture of the weighting scheme relative to other models.
 - Response: *Thank you for this comment. Those targets specifically imply there is a seasonality to the disease being forecast. Forecasting for specific seasonal dynamics would require different component models – ones that can predict peaks, not just short-term trends. EpiFFORMA should be interpreted as a general ensembling strategy – using a LightGBM and synthetic data to pre-learn disease dynamics – and should therefore allow one to capture seasonal dynamics, but this would require a different implementation where the component models are specifically designed to capture quantities such as peak trend. For many of the diseases investigated in this paper, the quantities peak week and incidence at peak week are not meaningful. Take COVID as an example. The notion of peak week is meaningful if one is restricted to a single outbreak/wave, but not meaningful otherwise. An implementation of epiFFORMA could be developed for those forecasting targets, but the component models in this paper would likely be inappropriate for this task.*
3. I thought it was an interesting choice to only train the model on synthetic outbreaks rather than any real outbreaks. Given your central argument is that the tool is disease agnostic, you could just include outbreaks from all different types of settings and pathogens. In that sense, you could additionally be accounting for (or at least considering) the differences in reporting and disease surveillance, which in the current method of synthetic outbreaks, is not accounted for (or at least it wasn't clear to me how it was accounted for). Assuming you continue forward with the current method of training the models only on synthetic data, I think it warrants a bit more rationale and discussion as to why this choice was made (presumably related to the number of time series available to train the model on, but I am not sure).
 - Response: *Thank you for bringing up this exciting point, which is a topic that holds a strong interest for our*

team. Training disease forecasting models (or, in this instance, ensembling meta-models) on cross-pathogen or cross-surveillance system data is an emerging interest in the literature, especially in the context of the “throw more data at it” mentality brought forth by advances in AI. See, for instance, the Flusion model [Ray et al., 2025], which placed 1st in the 2023-24 FluSight competition [Centers for Disease Control and Prevention (CDC), 2025], and several other recent papers [Coelho et al., 2020, Rodríguez et al., 2021, Chen et al., 2022].

The short answer to this comment is that we have another paper that specifically addresses the cross-disease and cross-surveillance system question. “Beyond Equal Weights” discusses an ensembling technique that is meant to perform well even if the behavior of an emerging pathogen behaves differently than any historical pathogen. We touch on this point with the UMAP analysis in the supplement; the synthetic data appears to reasonably “cover” several diseases that all behave very differently. Ideally, we would post a pre-print of the cross-disease paper and reference it in this paper, but the lead author on that paper is on leave until early 2026. We believe that this upcoming paper would make a good Nature Communications article.

To provide more rationale and discussion, we have expanded upon the question of real data in the discussion. Here, we argue more for the value of focusing on synthetic data in this paper, and mention that expanding these ideas to both synthetic and real data is explored in an upcoming paper.

4. And please comment further how reporting (is it time-varying?) is accounted for in the synthetic outbreaks.
 - Response: *Thank you for this comment. We do not do anything with that in the synthetic examples, though we are interested by the idea, especially since further research into the synthetic data library is an ongoing research topic for our team. Noise and bias in the observed data is represented in the synthetic data by adding in bias (to the rollercoaster trends) and noise into the synthetic data. The fact that the noise/bias can change over time in real data for the same geography/disease is addressed by the rolling window that is intrinsic to epiFFORMA. EpiFFORMA computes summaries of recent data: if noise and bias are changing over time in real data, those changes will be captured and will fall out of epiFFORMA’s summary window allowing epiFFORMA to naturally adapt to changing noise/bias in the reporting. This being said, dynamics caused by reported delays are not currently represented in the synthetic data. This comment has been added to the Supplementary Materials.*

Minor Comments

1. In the abstract, ILI has not yet been defined and may not be a well-known term to folks outside of infectious disease modeling. (also in Table 1, please define ILI)
 - Response: *Thank you for pointing this out. We write out Influenza-like Illness in both of these cases.*
2. ‘Dengue’ does not need to be capitalized
 - Response: *Thank you – this and the other disease names are no longer capitalized.*
3. Page 2, ‘forecaster’ should be forecast, I think.
 - Response: *Thank you, this has been corrected.*
4. Page 2, third paragraph – this situation was also observed with Zika virus (<https://www.nature.com/articles/s41467-021-25695-0>)
 - Response: *Thank you, this is a great addition! We’ve added mention of it to the article.*
5. The introduction reads very nicely
 - Response: *Thank you!*
6. Page 3, the data truncation process is not clear, please add more details
 - Response: *Thank you for catching this ambiguity. The word truncation has been removed and we are now more specific with what we meant here.*
7. Page 4, you refer to the top and bottom of Figure 1, I think you mean left and right? In any case, likely would make more sense to add A and B designations so you can be more precise in the reference to the figure
 - Response: *Thank you for catching this. (a) and (b) references have been added, and the text has been fixed.*
8. Figure 2 – why did the mirror model receive no weight? (this is brought up on page 10, but there should be a footnote in this figure to clarify)
 - Response: *Thank you for this comment. The information has been added to the caption of Figure 2.*

9. Figure 2 – I would suggest changing the y-axis label to something more informative... maybe “Relative error estimate” or something
 - Response: *Thank you, this has been added.*
10. Page 7 – I don’t think you’ve defined ‘UQ’
 - Response: *Thank you – we had not. We now write out ‘uncertainty quantification’ instead of using the abbreviation.*
11. Figure 4 – Please define GBM in the caption
 - Response: *Thank you, this has been added.*
12. It is useful to include the underlying intuition section. Consider adding a similar description for what the order represents.
 - Response: *Thank you for this comment; we have added more intuition around the order to this section.*

3 Reviewer 2 Comments:

1. The result was not explained in terms of the mathematics and statistics underlying the forecasting process - I was left wondering “why does this method out-perform equal weights?” an explanation would allow the result to be understood in the context of the literature around the FCP, and could potentially advance the field.
 - Response: *Thank you for this comment. The reason epiFFORMA out-performs equal-weights is because the meta-model (the LGBM weight model) is able to identify winners and losers well. For instance, Figure 1 in the paper shows that the best individual model is MOA and the meta model is able to assign the largest weight to MOA. Similarly in Fig S1, we show that the meta model really is able to predict which component models will do best and worst and assign weights to those models accordingly. For example, 25% of the time epiFFORMA gives the most weight to the component model that does the best (an equal weight ensemble would always give 11% of the weight to every model (1 out of 9)), indicating the meta model can identify winners. On the opposite side, 39% of the time epiFFORMA gives the least amount of weight to the worst model (again, the equal weight ensemble would always give 11% of the weight to every model), indicating the meta model can identify losers.*
To analyze this further, we have performed a cluster analysis on the weights (in our case, vectors of length 9). The clusters that are roughly closest to equal weights contain roughly 40.27% of the weights produced by the epiFFORMA ensembling meta-model across every forecast application in this paper; roughly 59.73% of the time epiFFORMA produced a distinctly non-equal weights profile.
The unequal weight model is not always a better choice. What unlocks the FCP is having a meta model able to discriminate between when an equal weight model is best and when something else is likely to be better. Much of this comment has been added to the discussion section.
2. I suggest a making a major effort to demonstrate this method relative to state-of-the-art epidemic forecasting techniques, particularly if those use equal-weighting and a contribution can be made towards explaining the FCP in this context.
 - Response: *Thank you for this comment. We have a strong preference for focusing this paper on an ensembling strategy – not about whether an implementation of this strategy out-performs certain forecasting models on certain diseases. However, we do believe that epiFFORMA would benefit from a comparison to other ensembling techniques. The literature for adaptive, online weighting strategies that do not incorporate historical data is somewhat bare. The closest that we found was [McAndrew and Reich, 2021], but this approach requires each component model to give probabilistic forecasts. We came up with a few others, such as using a Kalman filter on the weights, or recursive least squares with a forgetting factor, and we have visualized the cumulative average MAE of each of these strategies for forecasting COVID-19 in California in Figure 1. Over the course of the pandemic, the epiFFORMA technique reaches a lower average MAE than equal weights and the several online ensembling techniques considered. This figure supports the observations made in Ray et al. [2023], and continues to show that epiFFORMA does better than equal weights. We have added more details on this comparison to the supplementary materials.*
3. The db appears to contain legacy or non publication-related code (e.g., stuff_for_castro) I recommend making an updated user-facing release. The Zenodo database was a single 3GB zip file which prevented me from downloading and testing any of the code. Also it did not appear to be linked to an active repository.
 - Response: *Thank you for catching this. The code on the Github should now be run-able from a fresh pull, although we should note that it has been written in such a way that it relies heavily on a slurm HPC scheme.*

Figure 1: Comparison of alternative ensembling techniques for each of the four horizons.

References

- Centers for Disease Control and Prevention (CDC). Flusight 2023–2024 evaluation, April 2025. URL <https://www.cdc.gov/flu-forecasting/evaluation/2023-2024-report.html>. National Center for Immunization and Respiratory Diseases (NCIRD). Accessed 2025-09-10.
- Jing Chen, Germán G. Creamer, and Yue Ning. Forecasting emerging pandemics with transfer learning and location-aware news analysis. In *2022 IEEE International Conference on Big Data (Big Data)*, pages 874–883, 2022. doi: 10.1109/BigData55660.2022.10020218.
- Flávio Codeço Coelho, Nicolaus Linneu de Holanda, and Beatriz Coimbra. Transfer learning applied to the forecast of mosquito-borne diseases. *medRxiv*, 2020. doi: 10.1101/2020.02.03.20020164. URL <https://www.medrxiv.org/content/early/2020/02/04/2020.02.03.20020164>.
- Thomas McAndrew and Nicholas G. Reich. Adaptively stacking ensembles for influenza forecasting. *Statistics in Medicine*, 40(30):6931–6952, 2021. doi: <https://doi.org/10.1002/sim.9219>. URL <https://onlinelibrary.wiley.com/doi/abs/10.1002/sim.9219>.
- Evan L. Ray, Logan C. Brooks, Jacob Bien, Matthew Biggerstaff, Nikos I. Bosse, Johannes Bracher, Estee Y. Cramer, Sebastian Funk, Aaron Gerding, Michael A. Johansson, Aaron Rumack, Yijin Wang, Martha Zorn, Ryan J. Tibshirani, and Nicholas G. Reich. Comparing trained and untrained probabilistic ensemble forecasts of covid-19 cases and deaths in the united states. *International Journal of Forecasting*, 39(3):1366–1383, 2023. ISSN 0169-2070. doi: <https://doi.org/10.1016/j.ijforecast.2022.06.005>. URL <https://www.sciencedirect.com/science/article/pii/S0169207022000966>.
- Evan L. Ray, Yijin Wang, Russell D. Wolfinger, and Nicholas G. Reich. Flusion: Integrating multiple data sources for accurate influenza predictions. *Epidemics*, 50:100810, 2025. ISSN 1755-4365. doi: <https://doi.org/10.1016/j.epidem.2024.100810>. URL <https://www.sciencedirect.com/science/article/pii/S1755436524000719>.
- Alexander Rodríguez, Nikhil Muralidhar, Bijaya Adhikari, Anika Tabassum, Naren Ramakrishnan, and B. Aditya Prakash. Steering a historical disease forecasting model under a pandemic: Case of flu and covid-19. *Proceedings of the AAAI Conference on Artificial Intelligence*, 35(6):4855–4863, May 2021. doi: 10.1609/aaai.v35i6.16618. URL <https://ojs.aaai.org/index.php/AAAI/article/view/16618>.

Responses to Reviewer Comments on Manuscript “A Disease-agnostic Approach to Ensemble Learning for Infectious Disease Forecasting” for Nature Communications

February 2026

1 Note to all Reviewers:

We would like to thank all the reviewers and the Editor for their diligent suggestions for our article. No further edits were required.